# A Layered Hybrid Oxide–Sulfide All-Solid-State Battery with Lithium Metal Anode

**Juliane Hüttl** [1,†]**, Nicolas Zapp** [1,†]🄳**, Saoto Tanikawa** [1]**, Kristian Nikolowski** [1]🄳**, Alexander Michaelis** [1,2] **and Henry Auer** [1,*]

1   Fraunhofer Institute for Ceramic Components and Systems, 01277 Dresden, Germany;
    juliane.huettl@ikts.fraunhofer.de (J.H.); nicolas.zapp@ikts.fraunhofer.de (N.Z.);
    saoto.tanikawa@outlook.com (S.T.); kristian.nikolowski@ikts.fraunhofer.de (K.N.);
    alexander.michaelis@ikts.fraunhofer.de (A.M.)
2   Institute of Materials Science, TU Dresden, 01062 Dresden, Germany
*   Correspondence: henry.auer@ikts.fraunhofer.de
†   These authors contributed equally to this work.

**Abstract:** Different classes of solid electrolytes for all-solid-state batteries (ASSB) are currently being investigated, with each of them suitable for a different ASSB concept. Their combination in hybrid battery cells enables the use of their individual benefits while mitigating their disadvantages. The cubic stuffed garnet $Li_7La_3Zr_2O_{12}$ (LLZO), for example, is stable in contact with metallic lithium but has only moderate ionic conductivity, whereas the thiophosphate $Li_{10}SnP_2S_{12}$ (LSPS) is processable using conventional battery manufacturing technologies and has an excellent lithium-ion conductivity but an inferior electrochemical stability. In this work, we, therefore, present a layered hybrid all-solid-state full-cell concept that accommodates a lithium metal anode, a $LiNi_{0.8}Co_{0.1}Mn_{0.1}O_2$-based composite cathode with an LSPS catholyte (LSPS/NCM811) and a sintered monolithic LLZO separator. The electrochemical stability of LLZO and LSPS at cathodic potentials (up to 4.2 V) was investigated via cyclic voltammetry in test cells, as well as by cycling half cells with LSPS or a mixed LSPS/LLZO catholyte. Furthermore, the pressure-dependency of the galvanostatic cycling of a Li ⎮ LLZO ⎮ LSPS/NCM811 full cell was investigated, as well as the according effect of the Li ⎮ LLZO interface in symmetric test cells. An operation pressure of 12.5 MPa was identified as the optimal value, which assures both sufficient inter-layer contact and impeded lithium penetration through the separator and cell short-circuiting.

**Keywords:** electrochemistry; hybrid battery; lithium anode; lithium-ion battery; LLZO; solid-state battery; sulfide; thiophosphate



## 1. Introduction

The electrification of the automotive market, in combination with an increasing share of renewable energy sources, makes high-performing stationary, as well as mobile energy storage solutions, essential. The most common energy storage technology for mobile devices are secondary batteries, namely lithium-ion batteries. To meet the growing demands of the market, especially the automotive sector, the energy density of battery cells needs to be enhanced. This can be carried out either via the incorporation of high-voltage cathodes, like $LiNi_{0.8}Co_{0.1}Mn_{0.1}O_2$ (NCM811), or a lithium metal anode [1–3]. Lithium anodes have a theoretical capacity of 3860 mAh $g^{-1}$, which exceeds the capacity of commonly used graphite electrodes almost 10 times. In state-of-the-art batteries with liquid electrolytes, lithium metal electrodes cannot be used due to the instability of the liquid electrolytes and the risk of dendrite formation leading to cell failure.

Through the incorporation of solid electrolytes into the battery, not only does the safety increase due to lower flammability and lower risk of thermal runaway [4,5] but the use of lithium metal as anode also seems possible. There are several classes of solid electrolyte

materials, which can be categorized as oxidic, sulfidic and polymeric. Every material or material class has specific advantages, as well as certain drawbacks. Although polymeric electrolytes are easy to process and, thus, already used in industrial applications [6,7], they only have low ionic conductivities and are therefore not the optimal choice to enhance the energy density or power density of batteries compared to liquid electrolyte systems. Oxidic and sulfidic electrolytes exhibit higher conductivities at room temperature.

The stability toward the lithium metal anode is, in turn, the main advantage of the oxidic electrolyte $Li_7La_3Zr_2O_{12}$ (LLZO). In fact, LLZO is not thermodynamically stable toward lithium, but the degradation energy is very low, and the instability potential is only 0.05 V vs. $Li/Li^+$ [8–10], which is why it can be operated in direct contact with metallic lithium electrodes [11–16]. Therefore, LLZO can be employed as a sintered ceramic separator for facilitating a lithium metal anode. Furthermore, porous LLZO scaffolds may act as a host structure for the strainless plating and stripping of lithium metal [17].

On the other hand, LLZO is less suitable as a catholyte. The maximum ionic conductivity in the range of $10^{-3}$ S cm$^{-1}$ at room temperature [18–20] is sufficiently high to enable LLZO for thin electrolyte sheets, but it might hinder ionic transport through the complex structure of the cathode. Even more problematic in terms of using LLZO as a catholyte is the need for high-temperature sintering as the densification step. At high temperatures, LLZO tends to react with cathode materials like $LiCoO_2$ (LCO) or $LiNi_xCo_yMn_{1-x-y}O_2$ (NCM) [16,21,22]. The reactions were observed starting from 600 °C, though some interdiffusion even begins at temperatures as low as 300 °C [23]. Thus, the necessary co-sintering step of LLZO with the active material leads to the decomposition of both components and their loss of functionality.

The sulfidic electrolytes, however, have much higher ionic conductivities and can be combined with cathode active materials at low temperatures in processes well established in conventional battery manufacturing [1,24,25], albeit the requirements for production facilities are higher than those for conventional battery materials. A prominent example is $Li_{10}GeP_2S_{12}$ (LGPS), a sulfidic solid electrolyte, whose ionic conductivity of 12 mS cm$^{-1}$ matches those of liquid electrolytes [26,27]. Due to their ductility, sulfidic electrolytes can be processed via cold pressing and techniques derived from the production of conventional electrodes [6,26,28]. To gain cost effectiveness, germanium was replaced by tin, and the resulting compound $Li_{10}SnP_2S_{12}$ (LSPS) only showed a slightly lower lithium-ion conductivity than LGPS and a homologous crystal structure [29,30]. The main drawback of LGPS (and sulfidic electrolytes in general) is its narrow electrochemical stability window. Theoretical calculations report the electrochemical stability of LGPS as only between 1.71 and 2.14 V vs. $Li/Li^+$ [8], and the instability toward metallic lithium was confirmed via thorough experimental investigations [31]. They also show an electrochemical reactivity toward layered transition metal oxides, which does not, however, impede their use as a catholyte at moderate temperatures [32,33]; they are, therefore, promising regarding easy fabrication of solid-state batteries and can be used as cathode components but cannot be used in direct contact with lithium metal anodes. Therefore, a layered hybrid battery concept with an LLZO separator and sulfide as the catholyte would combine the benefits of a metallic lithium anode and a high-performing cathode.

In general, hybrid battery systems have gained increasing attention during the last few years. A comprehensive overview of different sorts of hybrid systems is given by Weiss et al. [34]. Most approaches make use of the facile processability of polymers and aim to increase their stability or conductivity. The combination of sulfidic and oxidic electrolytes has mainly been investigated in powder mixtures so far. For $Li_3PS_4$ (LPS) [35,36], as well as $Li_6PS_5Cl$ [37], it was shown that there are no interphases or degradation products detectable via XRD when mixed with LLZO powder. In a more recent work of our group, the interface resistance between layers of LPS ($Li_7P_3S_{11}$) and LLZO was investigated, showing that the intrinsic transition resistance for Li ions was unmeasurably low [38].

In this study, we expand our studies of the LLZO/LSPS hybrid system by constructing a battery cell consisting of a metallic lithium anode, an LLZO separator and a cathode, with an LSPS catholyte and $LiNbO_3$-coated NCM811 (cNCM) as the cathode active material (CAM)

(Figure 1). Besides having a low interfacial resistance, the electrochemical stability of the electrolytes at cathodic potentials is crucial. We, therefore, investigate their stability via cyclic voltammetry, as well as by incorporating LLZO particles as catholytes into an LSPS-based battery cell. Then, we aim to prove the applicability of the layered hybrid battery concept by combining an LLZO separator with a sulfidic cathode and investigate the effect of the stack pressure on the cycling behavior. Please note that for a commercial cell, the thickness of the employed separator will require minimization to improve energy and power densities.

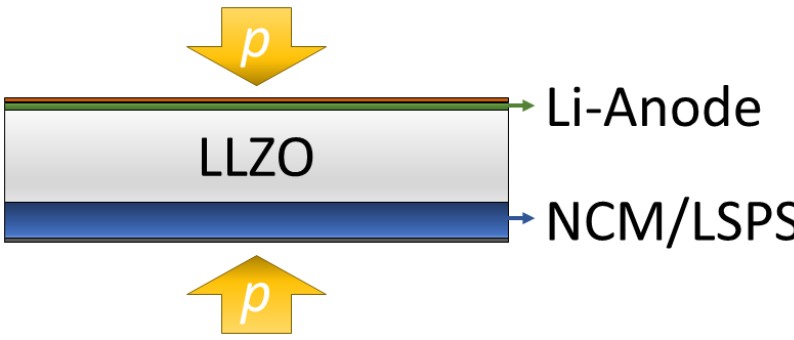

**Figure 1.** The pressurized layered hybrid cell concept investigated in this study contains a lithium-contacted sintered LLZO separator and a composite cathode, with LSPS as a catholyte and cNCM as an active material (Li | LLZO | LSPS/cNCM).

## 2. Materials and Methods

All materials were handled in an argon-filled (Linde, 99.999% purity) glovebox with monitored $H_2O$ and $O_2$ levels (<1 ppm). Sulfide samples were handled in a separated glove box to avoid cross-contamination. Aluminum-substituted LLZO powder was synthesized by annealing a mixture of $Li_2CO_3$ (VWR International, Darmstadt, Germany), $La_2O_3$ (Treibacher Industrie, Althofen, Germany), $Al_2O_3$ (Taimei Chemicals, Tokyo, Japan) and $ZrO_2$ (S. Goldmann, Bielefeld, Germany) for 6 h at 800 °C in air, followed by ball milling for 2 h (Pulverisette, Fritsch, Idar-Oberstein, Germany). The as-prepared powder was isostatically pressed with a maximum pressure of 700 MPa and subsequently sintered in MgO crucibles at a maximum temperature of 1240 °C for 4 h in air while being covered in an LLZO powder bed. Then, the sintered pellets were removed from the bed in inert atmosphere, cleaned, and weighed using an analytical balance, and the geometry was measured using a digital caliper (typically 10 mm diameter; 1.5–1.8 mm thickness; 4.7 g cm$^{-3}$ mass density). The surfaces of all LLZO pellets were roughened using a P120 SiC abrasive paper (WS flex 18 C; Hermes Schleifmittel, Dresden, Germany) to improve contacts to electrode materials and remove $Li_2CO_3$ impurities [38,39]. The by-impedance spectroscopy determined that the ionic conductivity of the LLZO pellets was $2.4 \cdot 10^{-4}$ S cm$^{-1}$ at 30.0 °C. Lithium electrodes and copper current collector sheets were attached to sintered LLZO pellets via hot pressing at 150 °C with a manual press using two heated plates; a force of 0.5 kN (corresponds to a pressure of 6.4 MPa) was applied for 1 min and, subsequently, 1.0 kN (12.7 MPa) for 3 min. We ensured that the lithium sheet covered the surface of the LLZO pellet after the treatment. For symmetrically contacted LLZO samples, the ceramics were fitted into poly-tetrafluoroethylene (PTFE) rings prior to contact to fit into the larger diameter of Swagelok-type cells (see below).

Electrochemical measurements were either conducted in Hohsen-type (Hohsen Corp., Osaka, Japan; 10 mm inner diameter) or Swagelok-type (Swagelok Company, Solon, OH, USA; 12.7 mm inner diameter) test cells. During assembly, LSPS-containing samples were cold compressed using a stationary hydraulic press. Subsequently, stack pressures in Hohsen-type cells were applied using a torque wrench (the application of 1.0, 5.0, 10.0, and 20.0 Nm corresponded to 2.5, 12.5, 25.0, and 50.0 MPa), whereas for Swagelok-type cells, a pneumatic vice with in-built manometer was used. The cells were placed in temperature-controlled cabinets and electrically attached to a BioLogic VMP3 potentiostat that was

operated via the software EC-Lab version 11.43 (both: BioLogic Science Instruments, Seyssinet-Pariset, France). Before starting the experiments, full cells and symmetric cells were temperature equilibrated for 24 h and 3 h, respectively, at open-circuit voltage (OCV).

For experiments with LSPS-containing separators, lithium–indium electrodes were attached by pressing a lithium chip with a 6 mm diameter and an indium chip with 10 mm diameter onto the solid electrolyte at 0.5 kN (75 MPa) for 5 s.

For cyclic voltammetry (CV) measurements, 97.0 mg LSPS (NEI Corporation, Somerset, NJ, USA; 95%, ionic conductivity of $2.0 \times 10^{-3}$ S cm$^{-1}$ at 30 °C; 'pure-LSPS-cell') or 67.9 mg LSPS and 29.1 mg LLZO powder ('LSPS/LLZO-cell') was mixed with 3.0 mg carbon (Super C65, Timcal Graphite and Carbon, Bodio, Switzerland) and thereafter placed on 200 mg LSPS in a Hohsen-type test cell before cold compression at 2 kN (250 MPa) for 1 min. Afterwards, Li-In was attached on the opposite side via the procedure described above, resulting in an asymmetrical semi-blocking cell setup Li-In | LSPS | LSPS/C | steel resp. Li-In | LSPS | LSPS/LLZO/C | steel. The measurements were conducted at 50 MPa and 60 °C between 2.0 and 4.0 V vs. Li/Li$^+$ with a scan rate of 0.1 mV s$^{-1}$. For the pure-LLZO-cell, a mixture of 97.0 mg LLZO powder and 3.0 mg carbon was pressed on a sintered LLZO pellet before attaching a Li-In electrode on the opposite side (Li-In | LLZO | LLZO/C | steel). CV was measured at 12.5 MPa and in otherwise identical conditions to those of the other cells. Estimated potentials vs. Li/Li$^+$ were calculated from the observed cell voltage by adding the potential of Li-In vs. Li (+0.62 V).

Cathodes for full cells were manufactured by mixing 69 mg LiNbO$_3$-coated LiNi$_{0.8}$Mn$_{0.1}$Co$_{0.1}$O$_2$ powder (1 wt.-% LiNbO$_3$; NEI Corporation) with 31 mg LSPS via pestle and mortar. Cathodes containing LSPS/LLZO-mixtures as catholyte were produced using 69 mg of cNCM, 21.7 mg of LSPS and 9.3 mg of LLZO powder. For cells with a LSPS separator, 20 mg of the cathode mixture (total capacity: 2.48 mAh; areal capacity: 3.18 mAh cm$^{-2}$) was placed on 100 mg of LSPS and compressed at 250 MPa for 1 min before attaching a Li-In anode on the opposite side. For the assembly of full cells with sintered LLZO separator, 10 mg of a cNCM/LSPS cathode mixture (1.24 mAh, 1.59 mAh cm$^{-2}$) was placed on a sintered and lithium-contacted LLZO pellet while ensuring that none of the cathode powder penetrated the small space between the inner ring of the Hohsen-cell and the LLZO pellet, which led to a short circuit. An aluminum sheet with a 10 mm diameter was added before compressing the cell at 50 MPa for 3 min.

For each full cell, the constant current constant voltage (CCCV) charging and CC discharging cycles were performed. Cells with lithium anodes were cycled between 3.0 and 4.2 V vs. Li/Li$^+$; those with Li-In anodes between 2.4 and 3.7 V vs. Li/Li$^+$. For full cells with a Li-In anode and LSPS- or LSPS/LLZO-based cathodes, CC rates varied from 0.02C (49.6 μA, 63.6 μA cm$^{-2}$) via 0.05C (124 μA, 159 μA cm$^{-2}$) and 0.1C (248 μA, 318 μA cm$^{-2}$) to 0.05C, whereas each rate was repeated once (8 cycles in total). During the constant voltage charging step, the current was limited to 24.8 μA. For hybrid layered full cells, CC rates varied from 0.01C (12.4 μA, 15.9 μA cm$^{-2}$) via 0.02C (24.8 μA, 31.8 μA cm$^{-2}$) and 0.04C (49.6 μA, 63.6 μA cm$^{-2}$) to 0.1C (124 μA, 159 μA cm$^{-2}$), repeating each rate twice. Subsequently, 10 cycles with 0.01C rates were applied (22 cycles in total). A constant voltage charging limit of 12.4 μA was used.

The pressure-dependent lithium stripping and plating behavior was investigated on symmetrically lithium-contacted LLZO samples. They were then investigated via direct current polarization tests under pressures varying from 2.5 to 37.5 MPa in Swagelok-type cells. Galvanostatic experiments were performed at 10.0 μA (12.7 μA cm$^{-2}$, which is close to the 15.9 μA cm$^{-2}$ applied in 0.01C charging/discharging experiments) and 30.0 °C via a 20 h per polarization cycle. Subsequently, the current was gradually increased to 200 μA (255 μA cm$^{-2}$) in 10.0 μA steps and thereafter to 1.00 mA (12.7 μA cm$^{-2}$) in 100 μA steps, whereas each current step was applied for 30 min before changing polarity. Potentiostatic electrochemical impedance spectroscopy (PEIS) was performed with a voltage amplitude of 25 mV and frequencies ranging from 1 MHz to 100 mHz, recording ten data points per frequency decade.

## 3. Results and Discussion

The chemical stability between LLZO and sulfidic electrolytes was shown in several works, for example, via XRD measurements [35–37]. Recently, our group also demonstrated the low interface resistance between LLZO and LPS [38]. To combine an LLZO separator with a sulfidic cathode, the electrolyte materials need to not only be stable in direct contact but also at potentials between 3 and 4.5 V vs. Li/Li$^+$. To investigate the electrochemical stability at these potentials, cyclic voltammetry experiments were conducted on LSPS and LLZO separately and compared to an LSPS-LLZO mixture. To enhance the reactive area, the powders were additionally mixed with carbon to form a 3D electronic-conducting network.

The CV curve of LSPS shows two pronounced degradation peaks in the first cycle with maximum intensities at 3.1 and 3.4 V (cf. Figure 2), which vanish by the second cycle. These can probably be attributed to the two-step oxidation of LSPS under the formation of elemental sulfur, i.e., $Li_{10}SnP_2S_{12} \rightarrow 2\,Li_3PS_4 + SnS_2 + 2\,S^0 + 4\,Li^+ + 4\,e^-$ (3.1 V) and $2\,Li_3PS_4 \rightarrow P_2S_5 + 3\,S^0 + 6\,Li^+ + 6\,e^-$ (3.4 V), analogous to observations made for LGPS and LPS [8,40]. LLZO does not show a significant peak in the applied voltage range, in accordance with the previous literature studies [12,15]. The curve of the mixture of LLZO and LSPS is analogous to that of the pure LSPS, with three additional peaks at 2.5, 4.0 and 4.4 V, which probably result from a reaction between LLZO and LSPS in the first cycle. Their intensity drops to zero by the second cycle, except for at 4.4 V, which remains significant for at least 7 cycles.

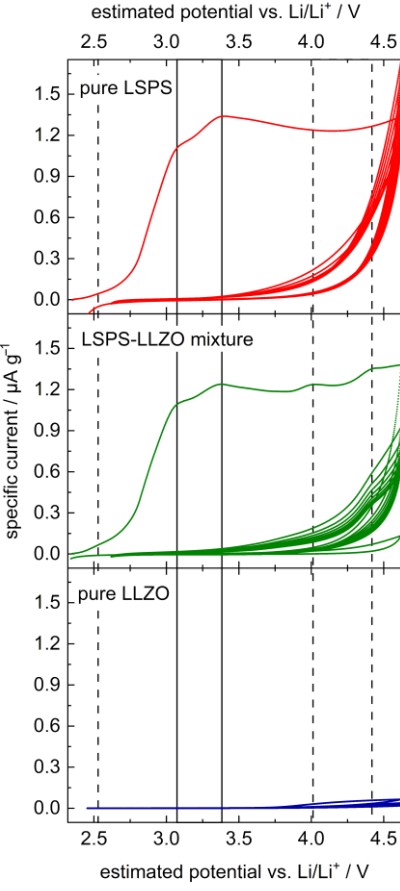

**Figure 2.** Cyclic voltammetry of an LSPS/LLZO powder mixture (**middle**, green) in comparison to pure LSPS (**top**, red) and LLZO (**bottom**, blue). Samples were mixed with carbon black to increase the reactive surface. Vertical lines highlight peak maxima at 2.5, 3.1, 3.4, 4.0 and 4.4 V vs. Li/Li$^+$; solid lines denote peaks observed in pure LSPS, and dotted lines denote additional peaks in the LSPS/LLZO sample.

To study whether the electrochemical reactions linked to these additional signals affect the cycling performance of an LSPS-based battery, LLZO powder was incorporated into an LSPS-cNCM cathode. Two cells containing a Li-In anode, an LSPS separator and an cNCM cathode, either with LSPS or an LSPS/LLZO mixture as the catholyte, were cycled with different C-rates at 60 °C (Figure 3). Both cells show stable cycling behaviors with similar progression and relative discharge capacities of 69.1% (LSPS) and 70.7% (LSPS/LLZO) in the first cycle. Differences can be observed in the ohmic drop, which are both more pronounced for the sample without LLZO, while the reduction in capacity upon cycling is higher for the LLZO-containing cell. The latter observation might result from the reactions observed in the CV experiment and reduce the long-term stability of a hybrid LSPS/LLZO cell. Additionally, these reactions affect the cell's internal resistance, which is higher for the LLZO-containing specimen (Figure A1), and, thus, explain the slightly inferior capacity retention upon cycling. Given the relatively minor scale of this phenomenon, the overall stability of the system can still be considered to be assured when operated at cathodic potentials.

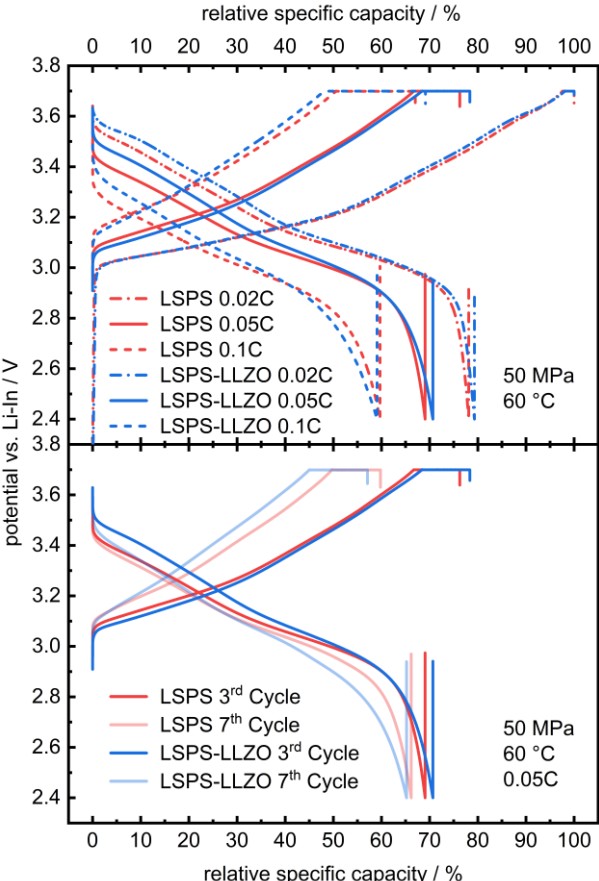

**Figure 3.** Selected charge and discharge curves of a full cell with an LSPS- (red) and LSPS/LLZO-based catholyte (blue; 30 wt.-% LLZO in the catholyte) and LSPS separator at 60.0 °C and 50 MPa stack pressure. The relative specific capacity was referenced to the first maximum charge capacity at 0.02C. The corresponding PEIS measurements are provided in Appendix A (Figure A1).

In a layered hybrid oxide–sulfide cell (Figure 1), the stack pressure is an important performance factor, since it provides inter-layer contact [38] and improves the performance of the composite cathode by counteracting internal de-contacting due to CAM breathing [32,41–43]. Similarly, it also affects the performance of the lithium anode [23,43–45]. Therefore, its effect on the Li-LLZO interface was investigated using symmetric Li | LLZO | Li test cells, as well as on a layered hybrid full cell.

Upon the long-term application of low charge densities in the latter, the potentials remained constant at 2.5 and 12.5 MPa and dropped at higher pressures after 5 *resp.* 3 h (Figure 4). Additionally, the critical current density of the cell decreases upon increasing the pressure from 382 µA cm$^{-2}$ at 12.5 MPa to 229 µA cm$^{-2}$ at 25 MPa (Figure 5). The corresponding impedance spectra show a significant drop in the cell resistance, which can be explained by the promoted formation of lithium dendrites in LLZO at higher pressures (Figures A2 and A3) [18,46,47]. The specimen's critical current density could further be improved by optimizing the lithium-LLZO interface [48] and the LLZO's microstructure [49], which would, however, probably not affect the observed pressure dependence.

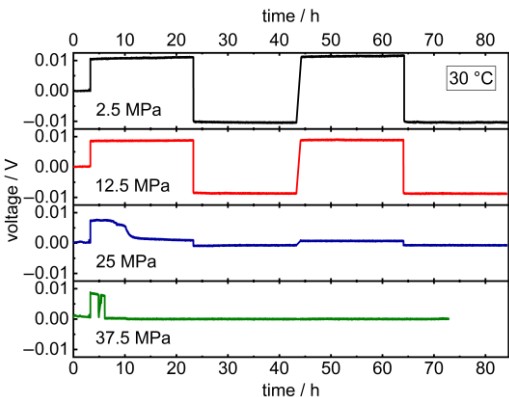

**Figure 4.** Galvanostatic lithium plating and stripping cycling profile of Li | LLZO | Li symmetric cells at 2.5, 12.5, 25 and 37.5 MPa stack pressure and 10 µA (12.7 µA cm$^{-2}$) at 30.0 °C. Corresponding PEIS data prior to and after cycling are provided in Appendix A (Figure A2).

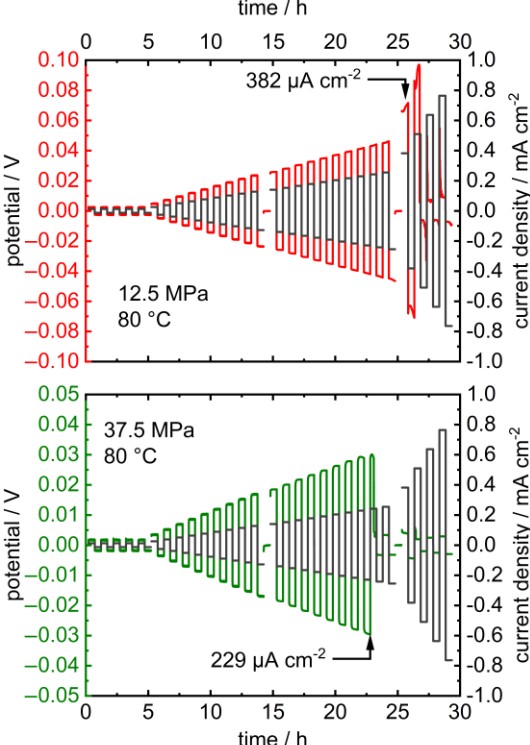

**Figure 5.** Stripping and plating experiments of symmetric Li | LLZO | Li cells with increasing current density at 12.5 MPa (**top**) and 37.5 MPa (**bottom**) and 80.0 °C (black lines: current density; red/green lines: potential). Please note the different scaling of the potential axis. The corresponding PEIS data before and after cycling are provided in Appendix A (Figure A3).

Similar results are observed upon cycling of layered hybrid Li | LLZO | LSPS/cNCM full cells: At 12.5 MPa, the cell can be charged and discharged (Figure 6; specific charge and discharge capacities: 160/107 mAh $g_{AM}^{-1}$). At higher stack pressures, the cell reaches medium potential levels, whose values decrease with increasing pressure (25 MPa: 3.8 V, 37.5 MPa: 3.6 V, 50 MPa: 3.5 V) while current is still flowing, i.e., the cell shorts [43]. From $p \geq 37.5$ MPa, potential drops upon charging are also observed. In combination with the results from the cycling of symmetric Li | LLZO | Li cells shown above, this observation is probably due to the penetration of lithium during cycling, which shortens the cell and prevents further charging. This is supported by the PEIS measurements before charging (Figure A4), after charging (Figure A5) and after discharging (Figure A6). The resistance of the cells at 25 MPa, 37.5 MPa and 50 MPa significantly drops, though not to zero. The cell is, thus, probably shortened by a number of lithium dendrites that penetrate the LLZO pellet.

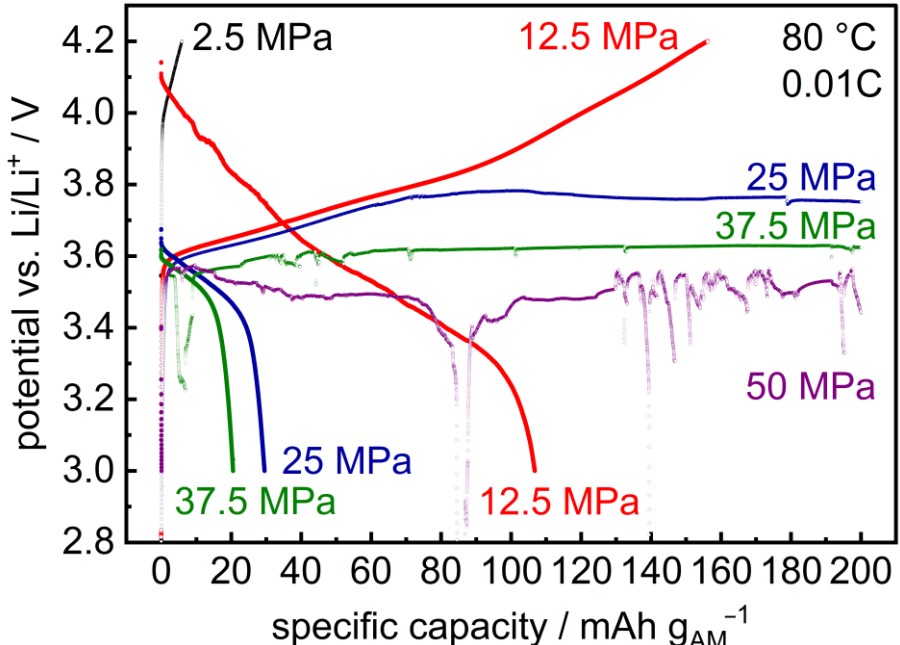

**Figure 6.** Initial charge and discharge curves of layered hybrid full cells Li | LLZO | LSPS/cNCM at different stack pressures (80.0 °C, 0.01C). The corresponding PEIS measurements are provided in Appendix A (Figures A4–A6).

At lower pressures (2.5 MPa), the cell can be charged (6 mAh $g_{AM}^{-1}$) but not significantly discharged, which is probably due to insufficient inter-layer contact between the LLZO separator and composite cathode. This is substantiated by the significant decrease in the cell's internal resistance upon increasing the stack pressure to $p \geq 12.5$ MPa (Figure A1).

At 12.5 MPa, the stack pressure hence provides sufficient inter-layer contact and still prevents the penetration of lithium through the separator. Therefore, this pressure was selected for the further cycling of a layered hybrid full cell at different rates (Figure 7). Due to the large thickness of the employed LLZO pellet (1.75 mm) and the resulting large internal resistance (Figure A7), a low rate of 0.01C was chosen at the beginning.

In total, 22 cycles could be performed, and, thus, this is to our knowledge the first time, that the concept of a multilayer setup with LLZO separator and sulfidic cathode was shown. However, the cell shows both a significant capacity loss over time, as well as low rate capability. Whereas at the beginning, a 126 mAh $g_{AM}^{-1}$ (70%) discharge capacity was obtained, it was 15 mAh $g_{AM}^{-1}$ (8%) in the 22nd cycle, and higher C-rates showed a significant drop in discharge capacities, which can be attributed to the already mentioned large internal resistance. These drawbacks probably result both from the setup with a manually mixed cathode with unoptimized morphology [41,42] and, most importantly,

the thick LLZO pellet, which caused a cycling behavior limited by the migration of Li-ions through the cell with a low power density. Additionally, the pressure-dependent performance of a thiophosphate-based composite cathode might require further adaption of the stack pressure. This is illustrated by the great increase in cell resistance upon cycling due to AM-breathing-induced internal decontacting (Figure A7). We are optimistic that further optimization of the components themselves and the setup, including the contact homogeneity at the interfaces, will increase the applicability of this cell concept. Due to the processability of LSPS at room temperature, this concept is a promising way to take advantage of the stability of LLZO toward the lithium anode without the need for a sintered oxidic cathode.

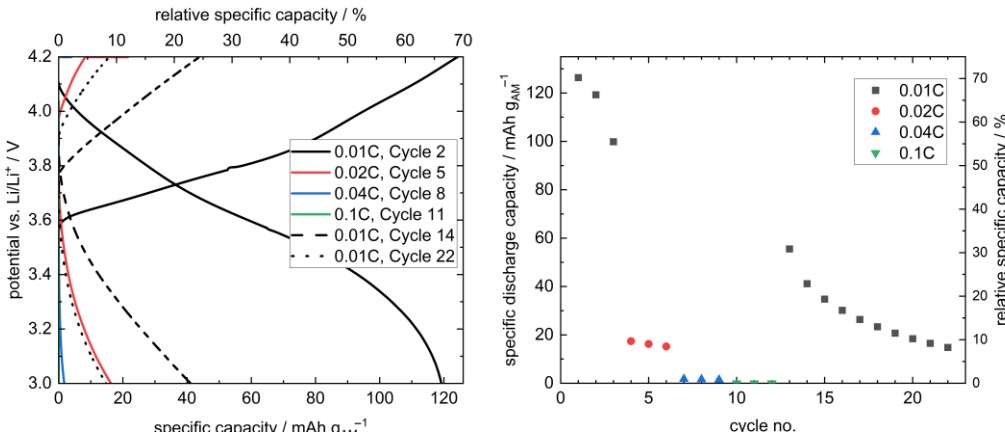

**Figure 7.** Selected charge/discharge curves (**left**) and cycle-dependent discharge capacities (**right**) of a sandwich-like hybrid full cell Li | LLZO | LSPS/cNCM at 80.0 °C and 12.5 MPa. A short temperature divergence resulted in the deviation of the charging curve in the 2nd cycle at 0.01C and 3.8 V vs. Li/Li$^+$. The relative specific capacity was calculated using the theoretical capacity of NCM811 at 4.2 V (180 mAh g$^{-1}$). The cycling curves at higher C-rates (0.04C (blue), 0.1C (green)) in the left plot are found at very small values and thus superimpose mostly with the axis in this representation. The corresponding PEIS measurements before and after cycling are provided in Appendix A (Figure A7).

## 4. Conclusions

In this study, the electrochemical compatibility between LSPS and LLZO was investigated, and a full cell with a lithium metal anode, sintered LLZO separator and LSPS-based NCM811 composite cathode was constructed. The mixture of LLZO and LSPS showed additional signals in the CV measurements compared to the single compounds, and the incorporation of LLZO powder into the catholyte only slightly reduced the performance of the cell. As these effects were of smaller magnitude, the overall stability of the combination of LLZO and LSPS was determined. The pressure-dependent performance of a sandwich-like hybrid cell concept was investigated, and 12.5 MPa was identified as the sweet spot between a high contact area of the LLZO and cathode, as well as the lower promotion area of lithium penetration. Finally, a full cell was constructed and cycled 22 times, which acts as proof of concept. The issues observed during this cycling, namely the capacity loss over time and low rate performance due to the high internal cell resistance, will be addressed in future investigations.

**Author Contributions:** Conceptualization: J.H., N.Z. and H.A.; Methodology: J.H., N.Z. and H.A.; Validation: J.H., N.Z. and S.T.; Formal analysis: J.H., N.Z. and S.T.; Investigation: J.H., N.Z. and S.T.; Resources: A.M.; Writing (Original Draft): J.H.; Writing (Review and Editing): N.Z.; Visualization: J.H., N.Z. and S.T.; Supervision: H.A., K.N. and A.M.; Project administration: H.A. and K.N.; Funding acquisition: H.A. and K.N. All authors have read and agreed to the published version of the manuscript.

**Funding:** This work was funded by the German Federal Ministry of Education and Research (BMBF, Germany) in the competence cluster for solid-state batteries in the Festbatt2-Oxide project (grant no. 03XP0434B).

**Data Availability Statement:** The data presented in this study are available on request from the corresponding author.

**Acknowledgments:** We thank Martin Drüe and Arno Ludwig Görne (Fraunhofer IKTS) for the synthesis of LLZO powder, as well as Jens Scholz and Kathrin Jungnickel (Fraunhofer IKTS) for help with the production of LLZO pellets.

**Conflicts of Interest:** The authors declare no conflict of interest.

## Appendix A

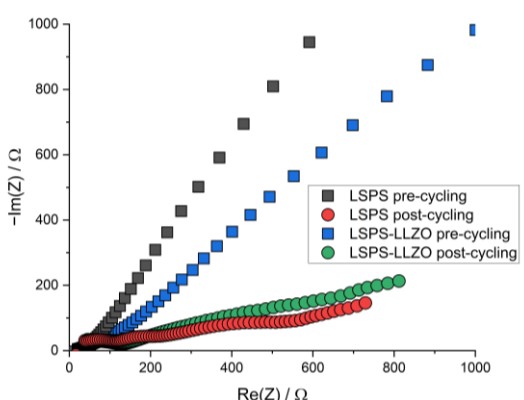

**Figure A1.** Nyquist plot of the PEIS spectra of LiIn | LSPS | LSPS/cNCM ('LSPS') and LiIn | LSPS | LSPS/LLZO/cNCM ('LSPS-LLZO') full cells before and after cycling at 60.0 °C.

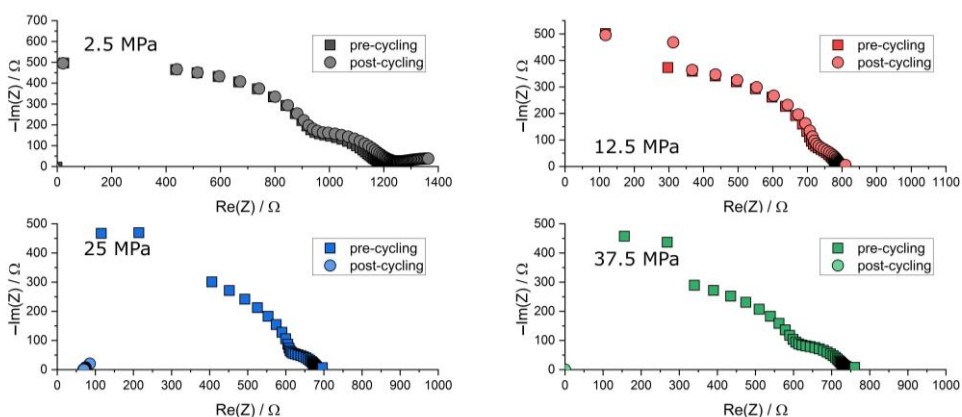

**Figure A2.** Nyquist plots of the PEIS spectra of Li | LLZO | Li specimen prior to and after chronopotentiometric cycling at different stack pressures and 30.0 °C. The data points at 37.5 MPa after cycling are situated at the origin due to short-circuiting.

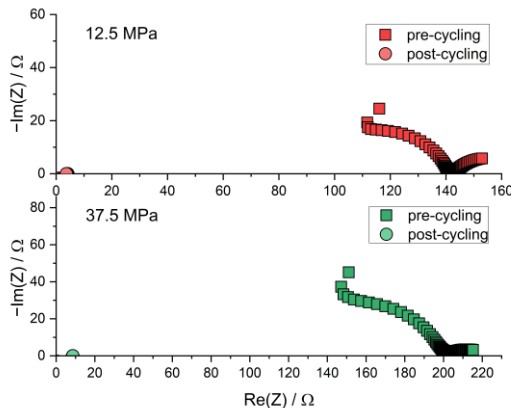

**Figure A3.** Nyquist plot of the PEIS spectra of Li ∣ LLZO ∣ Li specimen prior to and after chronopotentiometric cycling with increasing current densities at different stack pressures and 80.0 °C. The post-cycling data points are found near the origin due to short-circuiting.

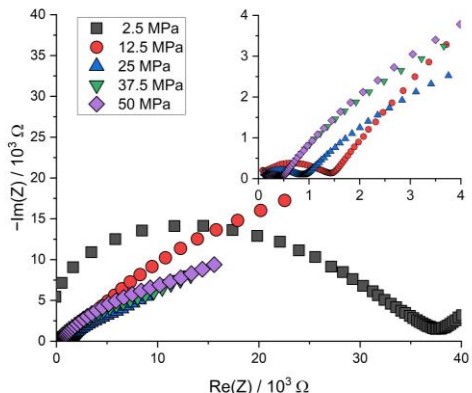

**Figure A4.** Nyquist plot of the PEIS spectra of Li ∣ LLZO ∣ LSPS/cNCM full cells at different stack pressures prior to cycling at 80.0 °C.

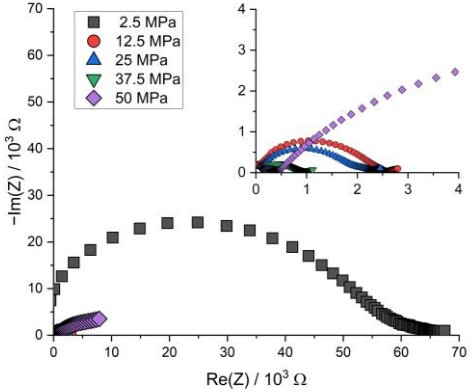

**Figure A5.** Nyquist plot of the PEIS spectra of Li ∣ LLZO ∣ LSPS/cNCM full cells at different stack pressures after charging at 80.0 °C.

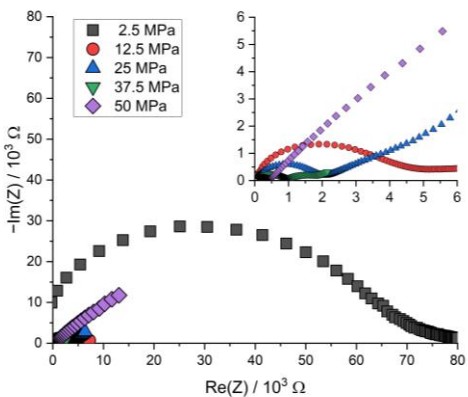

**Figure A6.** Nyquist plot of the PEIS spectra of Li | LLZO | LSPS/cNCM full cells at different stack pressures after discharging at 80.0 °C.

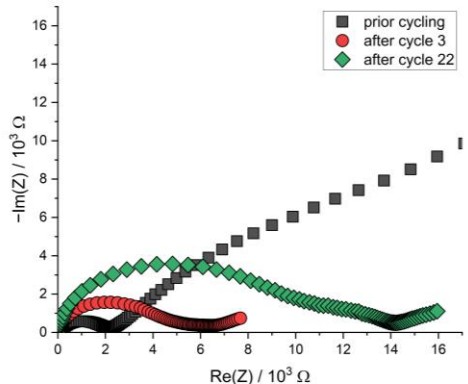

**Figure A7.** Nyquist plot of the PEIS spectra of the Li | LLZO | LSPS/cNCM full cell prior to cycling and after cycles 3 and 22 at 12.5 MPa and 80.0 °C.

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
