# Peer review of "A Layered Hybrid Oxide–Sulfide All-Solid-State Battery with Lithium Metal Anode"

_batteries, doi:10.3390/batteries9100507_

Round 1
Reviewer 1 Report
1. The authors present a layered hybrid all-solid-state full cell concept which accommodates a lithium metal anode, a LiNi0.8Co0.1Mn0.1O2 based composite cathode with LSPS catholyte (LSPS/NCM811) and a sintered monolithic LLZO separator. However, Li/LPSCl/NCM sulfide all-solid-state batteries were demonstrated stably cycle for hundreds of times (Materials Today Physics 32 (2023) 101009). The strength of authors provides system should be elaborated.
2. The reaction of LLZO and LSPS the first CV cycle. The exact reaction should be given corresponding to different voltage.
3. The passive layer of Li2CO3 on the surface of LLZO could attribute to the large internal resistance. The authors should avoid the influence of Li2CO3
4. The pressure also affect the performance of composite cathode, the authors should carefully estimate the performance pressure-dependent full cell performance.
5. Some related reference should be added: Batteries 2022, 8(10), 141; ACS Energy Lett. 2020, 5, 252−262; Chemical Engineering Journal, 2022, 429: 132343.
Reviewer 2 Report
This paper reports the electrochemical property of sulfide-based solid electrolyte LSPS as a catholyte for NMC cathode and oxide-based electrolyte LLZO as a separator. The effect for applying pressure on Li symmetric cell and full cell is systematically investigated. Although the strategy for using both LSPS catholyte and LLZO separator are interesting, the author should address to following comments and questions before publication.
In page 3:
Author should describe the density of LLZO.
In page 5:
In Figure 2, I recommend to revise the title of top horizontal axis to "Estimated potential vs. Li/Li+ is better expression." In addition, it is better to explain the details of oxidation reaction of LSPS at the potential above 2.0 V vs. Li-In alloy.
In page 6:
In Figure 3, We can observe the difference in ohmic drops in charge and discharge curves for full cells composed of LSPS or LSPS-LLZO catholyte and LSPS separator (Figure 3a). In addition, cycling stability for the full cell with LSPS catholyte is better than the full cell with LSPS-LLZO catholyte (Figure 3b). The authors should discuss about these difference. Have the author checked the EIS for these full cells?
In page 7:
For Figure 4, it is better to show the EIS data before and after galvanostatic cycling test for Li plating and stripping.
For Figure 5, although testing temperature is high (= 80 degree), obtained critical current density (CCD) for Li dendrite growth into LLZO is low reported in other papers (For example, more higher CCD is confirmed at room temperature in Batteries 8(10), 158, 2022). This is mainly caused by high applied pressure (12.5 MPa and 37.5 MPa)?
Page 9:
As shown in Figure 7, cycling performance for the full cell with LSPS catholyte and LLZO separator is very poor. Have the authors checked the change in EIS data during (or after) cycling test? What is the main reason for poor cycling performance?
Reviewer 3 Report
In this manuscript, an all solid state cell is designed with LLZO pellet as electrolyte and sulfidic catholyte mixed NMC cathode. And the stack pressure effect towards electrochemical performance is also discussed in this manuscript. However, either topic is not discussed thoroughly.
1. In order to make better conclusion for the first part, more experiment result is needed such as pure LLZO as catholyte, cycle stability comparison of different catholyte or CV of full cell. The author claimed the main different is the “ohmic drop”. Why sample with LLZO shows less ohmic drop? Since in introduction, author mentioned that, “LLZO is less suitable as catholyte” due to ionic conductivity? It is conflict with the data in Fig 3 that, catholyte with LLZO showed less ohmic drop. EIS or some other characterization is needed.
2. In stack pressure part, the experiment temperature is various. From 30C to 60C to 80C. Temperature is an important parameter, why not keep them constant? Why not perform the galvanostatic lithium plating and stripping at 60 or 80C as other experiment did?
3. Author claimed that, the abnormal voltage profile in Fig 6 is “probably due to” cell shorting. Why not test to confirm that instead of using the unsure word? You can do EIS to confirm shorting, you can do SEM of LLZO crosssection to confirm Li penetration. There are a lot of unsure words such as “probably”” might”, which is really not professional in an academic publication.
4. Is the cathode binder free? Because no binder is mentioned in experimental part.
5. Author claimed in introduction that “the main drawback of LSPS is its narrow electrochemical stably window.” Actually some other drawback of sulfidic catholyte is that they are hard to fabricate due to its critical requirement to fabrication environment. Its intrinsic instability also require cathode material to have coating on it.
Round 2
Reviewer 2 Report
The authors have addressed to all the questions or comments from the reviewers. I recommend its acceptance in present form.
Reviewer 3 Report
Author had addressed all the questions I have. The manuscript can be accepted as is.